# Self-correcting Reward Shaping via Language Models for Reinforcement Learning Agents in Games

**António Afonso[1,2], Iolanda Leite[2], Alessandro Sestini[1], Florian Fuchs[1], Konrad Tollmar[1], Linus Gisslén[1]**

amgcsa@hotmail.com, {asestini,ffuchs,ktollmar,lgisslen}@ea.com, iolanda@kth.com

[1]**SEED - Electronic Arts (EA), Stockholm, Sweden**
[2]**KTH Royal Institute of Technology, Stockholm, Sweden**

## Abstract

Reinforcement Learning (RL) in games has gained significant momentum in recent years, enabling the creation of different agent behaviors that can transform a player's gaming experience. However, deploying RL agents in production environments presents two key challenges: (1) designing an effective reward function typically requires an RL expert, and (2) when a game's content or mechanics are modified, previously tuned reward weights may no longer be optimal. Towards the latter challenge, we propose an automated approach for iteratively fine-tuning an RL agent's reward function weights, based on a user-defined language based behavioral goal. A Language Model (LM) proposes updated weights at each iteration based on this target behavior and a summary of performance statistics from prior training rounds. This closed-loop process allows the LM to self-correct and refine its output over time, producing increasingly aligned behavior without the need for manual reward engineering. We evaluate our approach in a racing task and show that it consistently improves agent performance across iterations. The LM-guided agents show a significant increase in performance from $9\%$ to $74\%$ success rate in just one iteration. We compare our LM-guided tuning against a human expert's manual weight design in the racing task: by the final iteration, the LM-tuned agent achieved an $80\%$ success rate, and completed laps in an average of $855$ time steps, a competitive performance against the expert-tuned agent's peak $94\%$ success, and $850$ time steps.

## 1 Introduction

Reinforcement Learning (RL) has the potential to significantly influence the gaming industry by enhancing both game development processes and player experiences (Jacob et al., 2020; Sestini et al., 2023). RL is a type of machine learning where agents learn through trial and error interactions with an environment, aiming to maximize a cumulative expected reward. It has been used for game testing (Zheng et al., 2019; Gillberg et al., 2023), game-AI (Torrado et al., 2018), and Procedural Content Generation (Khalifa et al., 2020; Zakaria et al., 2023). In recent years, research has increasingly focused on integrating human preferences through Preference-based Reinforcement Learning (PbRL) (Christiano et al., 2017), particularly in scenarios where explicit reward functions are difficult to specify. This approach enhances traditional methods by incorporating direct human feedback into the learning process. Unlike standard reinforcement learning, which depends on manually defined reward functions, PbRL leverages human preferences to automatically derive a reward function from

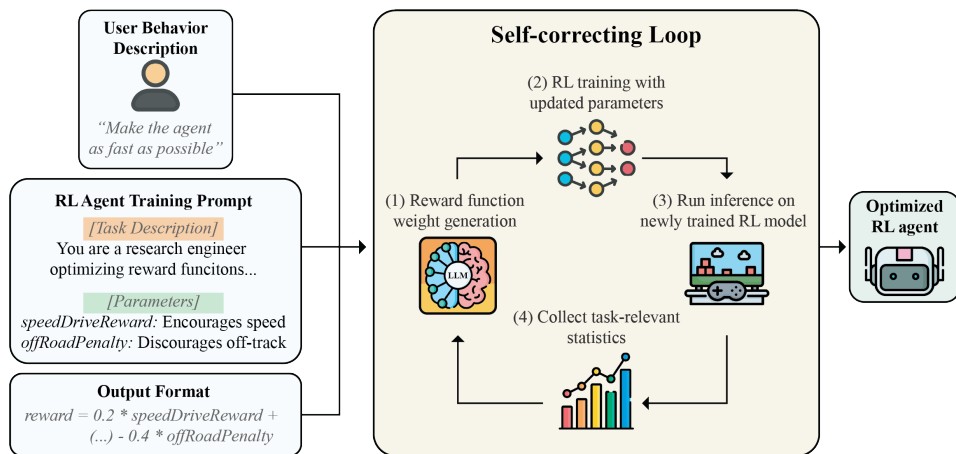

Figure 1: Overview of the self-correcting loop: following a user prompt (e.g., "*make the agent drive as fast as possible*"), alongside a pre-defined contextual prompt and example of an output format, the LM generates an initial set of reward parameter weights to fulfill the specified behavior (1). These parameters are used to train an RL agent (2). After training, inference runs are performed (3), and relevant task-related statistics (e.g., off-road occurrences, average speed) are collected (4). These statistics are then fed back into the LM, which iteratively refines the weights based on the observed outcomes. After $T$ iterations, the output of this pipeline is a trained RL model optimized for the user-specified behavior.

pairwise or groupwise preferences. These attributes allow people with little experience in RL, such as game designers and developers, to create RL with a more intuitive feedback loop. Thus, PbRL can potentially expand the usability of RL models, enabling game developers to tackle intricate tasks that are tough to define through conventional reward functions.

Despite these advances, two major challenges remain in production-grade game environments. First, classical reward engineering still demands RL expertise: crafting and tuning scalar reward weights often involves extensive trial-and-error and close collaboration between RL specialists and designers. Second, when a game's content or mechanics change – whether it be new levels, assets, or physics parameters – previously tuned weights may no longer be optimal, forcing repeated manual retuning.

Building on prior work that demonstrated LMs can produce reward functions comparable to a human expert (Kwon et al., 2023) and that iterative feedback can yield increasingly better-tuned reward scripts (Ma et al., 2023), we investigate whether a similar approach can be applied in a game-production context to shape specific designer-specified behaviors. Here, we evaluate the LM's ability to infer the mapping from abstract weighed reward components to observed performance metrics, and to progressively self-correct its weight proposals to converge on high-quality reward functions without direct expert intervention.

In this paper, we propose an automated, LM-guided loop for fine-tuning reward function weights in arbitrary game scenarios. At each iteration, the LM receives (1) a high-level description of the task and environment, (2) the full history of weight vectors $\{\mathbf{w}^{(0)}, \ldots, \mathbf{w}^{(i-1)}\}$, and (3) concise performance statistics from prior agent evaluations. The LM then proposes a new weight vector $\mathbf{w}^{(i)}$, which is used to train the RL agent and collect new statistics, closing the feedback loop. By incorporating this feedback loop, the LM becomes more adept at aligning the reward structures with desired game behaviors.

We evaluate this pipeline on a car racing task. Our findings demonstrate that this method significantly improves the LM's output quality, resulting in more effective reward functions for RL agents. Consequently, these refined reward functions contribute to better agent performance and potentially

create more engaging gameplay experiences, highlighting the potential of integrating LMs into game development to optimize AI-driven elements.

## 2 Related Work

RL has been increasingly applied to game development tasks such as automated testing, NPC behavior design, and procedural content generation. For instance, Sestini et al. (2023) propose a manual, imitation learning-based "patching" workflow where designers record new trajectories to correct undesirable in-game behaviors. While effective, such approaches still require substantial developer effort to identify, collect, and integrate corrective demonstrations. LMs have also been used to automate distinct aspects of reward design. Kwon et al. (2023) demonstrate that LMs can act as preference models to rank agent trajectories from high-level task descriptions, enabling reward learning without explicit reward functions. Other approaches – such as population-based training Jaderberg et al. (2017) – create cohorts of agents with varied reward parameters and select the best performers; however, they discard previous results rather than reusing them, making them computationally expensive.

PbRL methods incorporate human feedback directly into the reward model. FLORA explores the low intrinsic dimensionality of robotic reward functions to adapt behaviors to new human-style preferences with minimal parameter changes (Marta et al., 2025). This approach shows that compact, interpretable parameterizations can ease rapid adaptation, despite still relying on explicit human-in-the-loop tuning. Other work focuses on improving LM outputs via self-correction or code generation: Han et al. (2024) explores intrinsic self-correction in small LMs, requiring fine-tuning to iteratively refine textual outputs, while Chain-of-Thought prompting enhances zero-shot reasoning without additional training (Kojima et al., 2022). Klissarov et al. (2024) push this further by using LMs to generate Python code that assembles and trains sub-behavior modules for RL agents.

A closely related loop-based approach is EUREKA, which similarly employs an LM in a loop to improve game-related behaviors (Ma et al., 2023). However, the method differs from ours in that its objective is to discover entirely new skill modules via evolutionary code generation, whereas our method focuses on fine-tuning an existing reward function to achieve a specified behavior. Unlike EUREKA, which requires access to the raw environment source code, we operate solely on an abstract, human-readable summary of reward components using significantly less compute power.

## 3 Methodology

Our approach follows a self-correcting loop in which a Language Model (LM) iteratively adjusts the weights of a modular reward function to better align an RL agent's behavior with a user-defined objective. This can be viewed as a form of preference-based learning, where the user provides high-level behavioral preferences, and the LM tunes the reward parameters to elicit that behavior. The method operates in four key stages: (1) a reward structure with tunable weights is defined; (2) given an environment description and user preference (e.g., "*drive as fast as possible without going off track*"), the LM proposes an initial set of reward weights; (3) an RL agent is trained with these weights, and its performance is evaluated; and (4) performance statistics are summarized and passed back to the LM, which proposes new weights in response. In the following section, we further describe each of these steps. Figure 1 shows the full pipeline.

### 3.1 Self-Correcting Reward-Shaping Pipeline

We frame reward shaping as an iterative, closed-loop optimization over a decomposed, linear reward function of the form:

$$r_t \;=\; \sum_{k=1}^{K} w_k \, f_k(s_t, a_t) \,, \tag{1}$$

where:

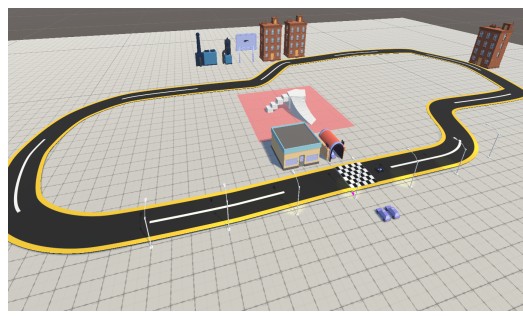
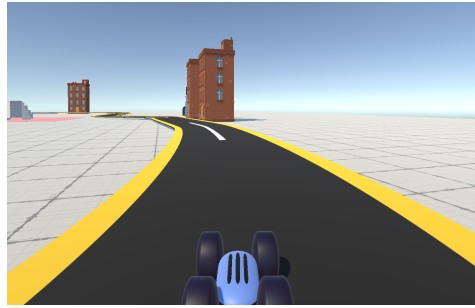

(a) Racing Environment: Bird-Eye view

(b) Racing Environment: First-Person view

Figure 2: The racing environment used in this paper. The driving agent is initially located ahead of the goal line, and has to drive around the racetrack until reaching the goal line again (checkers).

- $w_k$ is a scalar weight (e.g. `goalReachedReward`, `speedDriveReward`, as we will see in Section 4.1) that tunes the importance of that feature.

- $f_k(s_t, a_t)$ is a modular feature function evaluating one aspect of task-specific behaviors (e.g., number of time steps, current speed, distance to the center of the road etc.).

This abstraction separates *what* aspects of behavior are measured ($f_k$) from *how strongly* they are enforced ($w_k$). Our LM-guided loop (shown in Figure 1) therefore treats reward shaping as a problem of optimizing the weight vector $\mathbf{w} = (w_1, \dots, w_K)$ – where $K$ is the number of weights present in the environment – without altering the underlying feature implementations. This design not only generalizes across driving tasks, but also in other environments, by instantiating a respectively appropriate set of $f_k$, followed by the same iterative weight-tuning pipeline.

Our pipeline runs for $T = 5$ iterations. The choice for this reflects our empirical observation of diminishing performance beyond five iterations, and aligns with similar iterative approaches in the literature (Ma et al., 2023). Each of these iterations consists of:

1. **LM-guided weight proposal.** The pipeline first creates a prompt containing:

    - A brief description of the current environment.

    - The user's high-level objective (e.g. "drive as fast as possible without leaving the track").

    - The history of previous weight vectors $(\mathbf{w}^{(0)}, \dots, \mathbf{w}^{(i-1)})$ together with summary performance metrics (success rate, off-road percentage, average speed). Note that for $i = 0$, this input is empty.

    The Language Model takes this prompt and outputs a new weight vector $\mathbf{w}^{(i)}$.

2. **RL training and evaluation.** We train an RL agent to convergence under reward $r_t = \sum_k w_k^{(i)} f_k$. Once training completes, we run the policy for 50 evaluation episodes across 5 different seeds and collect relevant performance statistics that vary according to the environment used. More details regarding the configuration or hyperparameters can be found in Section C of the Appendix.

3. **Statistical feedback and refinement.** The newly collected metrics are fed back into the LM prompt for the next iteration. The LM refines its suggested weights $\mathbf{w}^{(i+1)}$ based on both the raw reward definitions (Eq. 1) and the history of performance data – enabling intrinsic self-correction without any model fine-tuning.

At the end of step 3, one iteration ends, and the next one restarts from step 1. After $T = 5$ iterations, the loop outputs a final, optimized RL policy plus a reward function with optimized weights. The particular set of feature functions, $f_k$, is defined per environment. We provide a full list of all components, $w_k$, in Section 4.1.

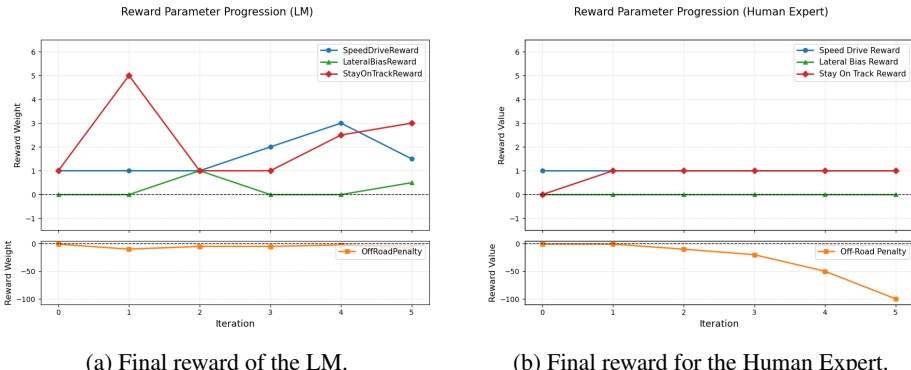

(a) Final reward of the LM.   (b) Final reward for the Human Expert.

Figure 3: Reward weight progression across iterations for the LM (left) and human expert (right). Each plot is split into two subplots to separate lower-magnitude parameters (top) from the `offRoadPenalty` (bottom).

## 4   Experiments

In order to test our hypothesis that LMs can self-correct their generated output in the form of a reward function, we perform experiments on a racing task in an environment proposed by Sestini et al. (2023). Figure 2 shows the environment. For all of the experiments conducted, we chose OpenAI's o3[1] language model, mainly due to its recent proven performance on dealing with large context windows and high-reasoning abilities (Ballon et al., 2025).

This environment simulates a racing scenario, common in games ranging from kart racers and driving simulators to first-person shooters featuring vehicle mechanics. In this experiment, the user-specified objective is to complete laps around a racetrack as quickly as possible – a classic problem in this domain. However, the challenge lies in the inherent trade-off between speed and control. As shown in the work by Fuchs et al. (2021), merely maximizing forward acceleration is not sufficient – agents that ignore safety constraints often veer off the track, resulting in failed episodes or poor lap completion. Effective driving thus requires balancing aggressive acceleration with precise trajectory control. While this particular setup aims for fast lap times, it's important to note that the broader framework allows for flexible behavior shaping, depending on the user's preferences. For instance, a designer could instead specify a cautious driving style, or propose lane adherence favoring the left vs. right side of the track. This scenario thus should not be viewed as a search for a single optimal policy, but as one example within a space of possible behaviors that a designer might want to evoke.

We evaluate agent behavior using three key metrics: average lap completion time, off-road incidence, and success rate (i.e., the proportion of episodes where the agent completes the lap without crashing or timing out). An effective reward function in this context is one that aligns agent behavior with the specified user intent – whether that means maximizing speed, prioritizing stability, or any other nuanced driving pattern.

### 4.1   Reward Function Components

The environment uses a decomposed reward function comprised of weighted sub-rewards, where each component promotes or discourages specific behaviors. These reward components, $f_k$, are the targets of optimization in our self-correcting loop and are adjusted iteratively through $w_k$ by the LM to better align with user-defined objectives. Below, we outline the set of reward components used across our task, including their purposes and how they influence agent behavior. For a full look at the LM prompt with more detailed reward descriptions, refer to Sections A and B of the Appendix.

---

[1] https://openai.com/index/introducing-o3-and-o4-mini/

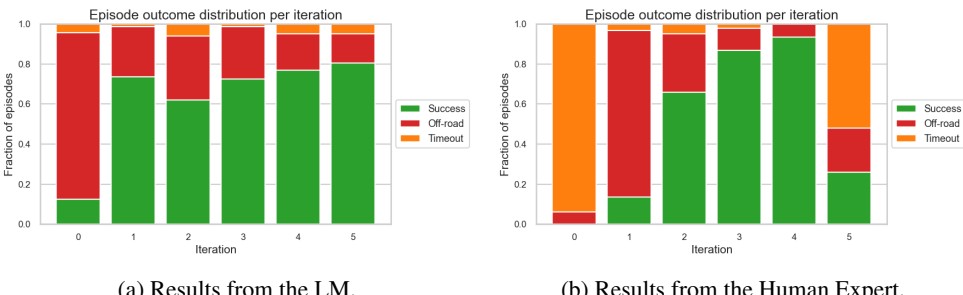

(a) Results from the LM.  (b) Results from the Human Expert.

Figure 4: Outcomes from post-training inference over five iterations for the LM (left) and Human Expert (right).

- **speedDriveReward**: Rewards the agent for maintaining high speeds and accelerating efficiently. This encourages an aggressive driving style.

- **offRoadPenalty**: Penalizes the agent for leaving the racetrack.

- **lateralBiasReward**: Encourages the agent to maintain a desired lateral alignment on the road (in relation to a central spline). This helps guide the agent toward preferred driving lanes and can be tuned to promote left/right side bias in multilane scenarios.

- **stayOnTrackReward**: Rewards alignment with a predefined spline or path, encouraging consistent trajectory following. It works in tandem with the lateral bias to reinforce smooth navigation around corners or along designated lanes.

Each of these components can be scaled up or down via their respective weights, and the iterative feedback loop seeks to optimize this weighting to match the target desired behavior.

## 5 Results

In this section, we report the results of our approach in the environment described in Section 4. To evaluate the LM's effectiveness in the iterative weight adjustment pipeline, we present a side-by-side comparison between an LM-driven loop and a human expert-driven loop within the same racing task. The objective in both cases was to train an agent to minimize lap time while avoiding off-road penalties and timeouts – posing a clear tradeoff between speed and control. Both the LM and the human expert operated under identical conditions: the feedback loop remained unchanged, with the only difference being the source of the updated reward weights. In each iteration, the agent's performance metrics (e.g., success rate, average speed) and reward weight history from previous iterations were compiled into a textual prompt. This prompt was then either sent to the LM or shown to the human expert, who returned a new reward weight vector for the next round. Neither the LM nor the human had access to videos, visualizations, or direct interaction with the environment – they both relied solely on the structured summaries provided in the prompt. Performance metrics were collected by running inference on each trained model over 50 evaluation episodes, repeated across five random seeds, for a total of 5 iterations. The results are visualized and compared in Figures 3–5 and summarized in Table 1.

Figure 3 presents the final weight configuration proposed by the LM and the human expert. Overall, the LM makes relatively small but consistent adjustments across iterations. This suggests a stable optimization process in which the model gradually learns to balance competing objectives. In contrast, the human expert makes more dramatic changes in the configuration per iteration, which leads to sharper performance shifts between iterations.

Performance outcomes over iterations are detailed in Figure 4 and Figure 5, with a full summary in Table 1. In the early iterations, the LM rapidly improves agent performance. Starting from a baseline success rate of just 12.4% (Table 1a), it jumps to 73.6% in the very first feedback iteration

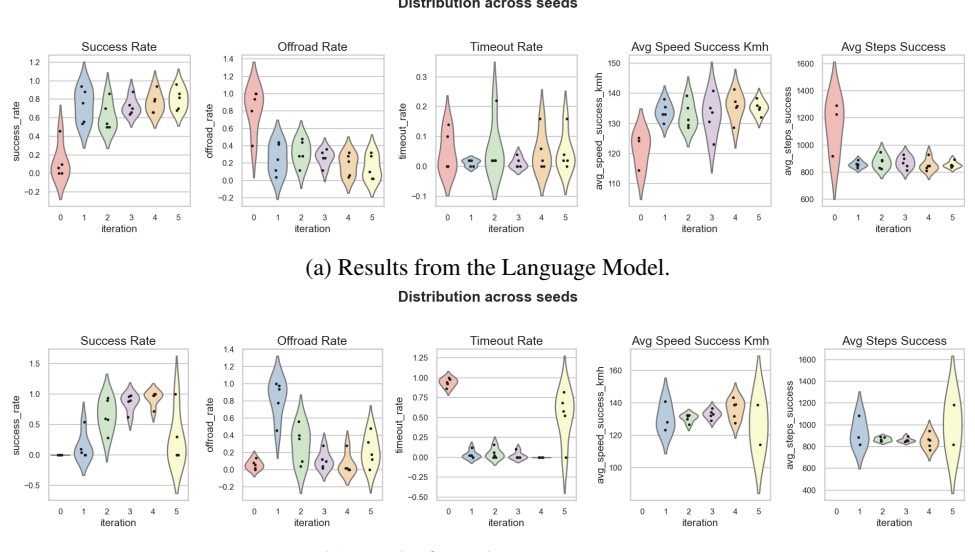

(a) Results from the Language Model.

(b) Results from the Human Expert.

Figure 5: Violin chart with various performance metrics from inference over the 5 iterations for the LM (top) and Human Expert (bottom).

and continues to improve steadily, peaking at 80.4% by iteration 5. At the same time, the LM reduces off-road occurrences from 83.2% to 14.8%, and increases average speed from 121.3 to 135.2 km/h, demonstrating an ability to balance multiple objectives without explicit reward engineering.

This early gain also suggests that even limited feedback – available only after the first iteration – plays a vital role in helping the LM reason over the reward space. While the LM starts from a simple, uninformed guess in iteration 0, it quickly produces competitive results once feedback is incorporated.

The human expert (Table 1b), in contrast, began with a 0.0% success rate and 94.0% timeouts in iteration 0, but rapidly improved performance through successive iterations. By iteration 4, the expert reached a peak success rate of 93.6%, with just 6.4% off-road behavior and 0% timeouts – outperforming the LM on most individual metrics. In iteration 5, however, performance dropped: success rate declined to 26.0%, timeouts increased to 52.0%, and off-road behavior worsened. While not necessarily indicative of a flawed reward design, this shift highlights the non-linear and occasionally brittle nature of manual tuning, where large adjustments may occasionally degrade performance.

The violin plots in Figure 5 help illustrate this contrast. The LM results (5a) show a narrowing distribution over time, reflecting an increasingly consistent agent behavior. The human expert metrics (5b) show more variation between iterations, particularly in speed and episode duration, suggesting a more exploratory tuning process.

Taken together, these results show us two different strategies that have successfully aligned the agent's behavior toward the initial user goal. The human expert, drawing on intuition and experience, was able to reach higher peak performance. The LM-driven approach, on the other hand, required no manual input and produced strong results early in the process, thanks in part to its ability to incorporate feedback iteratively. This makes it especially valuable in scenarios where fast iteration or reduced reliance on RL expertise is a priority, offering a practical and accessible pathway to reward tuning in game environments.

| Iteration | Success [%]↑ | Off-road [%]↓ | Timeout [%]↓ | Speed [km/h]↑ | Steps |
|---|---|---|---|---|---|
| 0 | $12.4\%^{+4.7}_{-3.5}$ | $83.2\%^{+4.4}_{-5.3}$ | $4.4\%^{+3.3}_{-1.9}$ | $121.3 \pm 6.0$ | $1147 \pm 200$ |
| 1 | $73.6\%^{+5.1}_{-5.8}$ | $25.2\%^{+5.7}_{-5.0}$ | $\mathbf{1.2\%}^{+2.3}_{-0.8}$ | $134.0 \pm 3.1$ | $855 \pm 22$ |
| 2 | $62.0\%^{+5.8}_{-6.2}$ | $32.0\%^{+6.0}_{-5.5}$ | $6.0\%^{+3.7}_{-2.3}$ | $132.7 \pm 4.4$ | $875 \pm 50$ |
| 3 | $72.4\%^{+5.2}_{-5.8}$ | $26.4\%^{+5.8}_{-5.1}$ | $1.2\%^{+2.3}_{-0.8}$ | $132.7 \pm 6.5$ | $871 \pm 45$ |
| 4 | $76.8\%^{+4.8}_{-5.6}$ | $18.4\%^{+5.3}_{-4.3}$ | $4.8\%^{+3.4}_{-2.0}$ | $\mathbf{135.7} \pm 4.6$ | $\mathbf{852} \pm 45$ |
| 5 | $\mathbf{80.4\%}^{+4.4}_{-5.4}$ | $\mathbf{14.8\%}^{+4.9}_{-3.9}$ | $4.8\%^{+3.4}_{-2.0}$ | $135.2 \pm 2.3$ | $855 \pm 22$ |

(a) Summary of results from the Language Model.

| Iteration | Success [%]↑ | Off-road [%]↓ | Timeout [%]↓ | Speed [km/h]↑ | Steps |
|---|---|---|---|---|---|
| 0 | $0.0\%^{+1.5}_{-0.0}$ | $\mathbf{6.0\%}^{+3.7}_{-2.3}$ | $94.0\%^{+2.3}_{-3.7}$ | – | – |
| 1 | $13.6\%^{+4.8}_{-3.7}$ | $83.2\%^{+4.1}_{-5.1}$ | $3.2\%^{+3.0}_{-1.6}$ | $130.8 \pm 9.1$ | $929 \pm 138$ |
| 2 | $66.0\%^{+5.6}_{-6.1}$ | $29.2\%^{+5.9}_{-5.3}$ | $4.8\%^{+3.4}_{-2.0}$ | $130.7 \pm 2.5$ | $867 \pm 19$ |
| 3 | $86.8\%^{+3.6}_{-4.8}$ | $11.2\%^{+4.5}_{-3.3}$ | $2.0\%^{+2.6}_{-1.1}$ | $133.1 \pm 2.8$ | $861 \pm 19$ |
| 4 | $\mathbf{93.6\%}^{+2.4}_{-3.7}$ | $6.4\%^{+3.7}_{-2.4}$ | $\mathbf{0.0\%}^{+1.5}_{-0.0}$ | $\mathbf{136.2} \pm 6.3$ | $\mathbf{850} \pm 67$ |
| 5 | $26.0\%^{+5.8}_{-5.0}$ | $22.0\%^{+5.5}_{-4.7}$ | $52.0\%^{+6.1}_{-6.2}$ | $126.7 \pm 17.3$ | $1002 \pm 258$ |

(b) Summary of results from the Human Expert.

Table 1: Key performance metrics per iteration for both (a) the LM and (b) the Human Expert. All rates are Wilson 95% confidence intervals; both speed and steps are presented with mean ± standard deviation. The best value for each metric is highlighted in bold.

# 6    Conclusion and Future Work

We proposed a method to automate reward shaping for RL agents using LMs in an iterative loop setup. While previous work has shown that LMs can generate plausible initial reward functions (Ma et al., 2023), our preliminary results demonstrated that incorporating feedback through an iterative refinement process substantially improves their output. Our results show that this automated approach can produce reward functions that are competitive with those crafted by a human expert, without requiring manual tuning at each step. This method offers benefits for game production: it enables automatic re-tuning of agents when game parameters change, and it lowers the barrier for non-experts to integrate RL agents via intuitive, high-level goals.

A current limitation of our approach is that the LM operates solely on textual prompts and scalar performance metrics, without access to the visual environment or the agent's behavioral trajectories. It is then up to the users to design which metrics should be reported. While this worked well in our experiments, it may miss subtle behavioral cues that would otherwise be perceivable in video or visual inspection. An interesting avenue for future work is then the integration of visual feedback. We conducted preliminary experiments using a Vision-Language Model to process single-frame observations of agent behavior. While the idea is worth investigating, this setup is not yet robust enough for practical reward shaping Faldor et al. (2025). Incorporating richer sequences or videos into the LM feedback loop remains a challenge, but it could significantly improve the model's ability to interpret agent behavior holistically. We leave this for future exploration as visual foundation models continue to improve. In parallel, future experiments should explore a broader range of environments to ensure generalizability – including established RL benchmarks and visually rich settings e.g. VizDoom (Kempka et al., 2016) – where user-defined behaviors may be more diverse and nuanced.

# A  RL Agent Reward-Function Task Description

In this section we show an example of the initial prompt we feed to the LM at the start of the pipeline described in Section 3.1. The prompt includes the preference of the designer, who describes the desired behavior.

---

**LM Prompt**

## Task Description

You are a research engineer at a gaming company, optimizing the reward function for a reinforcement learning (RL) agent in a Unity-based environment. The environment, trained using ML-Agents, contains car-driving RL agents that can:

• Move forward and backward

• Steer the vehicle to the left and right

You will receive a file called `rewardFunction.txt`, which will contain the reward signals and their respective scaling factors. Your main focus should be on adjusting the scaling factors, as they determine the strength of each reward signal. Note that the values in this file are neutral placeholders – they are not optimized and exist only to illustrate the file structure. **Do NOT** take inspiration from these values when tuning the parameters. When a parameter is not mentioned in the Problem section, **you MUST SET ITS VALUE TO 0**.

### Problem (User Prompt)

I have a driving scene where there is a racetrack and a driving RL agent that is trying to learn how to drive. I want to make the driving agent to **drive as fast as possible, without falling off the racetrack**. This means I want my agent to be able to consistently do laps on the racetrack, and doing it as fast as it can.

Your task:

1. Modify the reward function parameters to ensure the driving agent is going **as fast as possible**.

2. Modify the reward function parameters to ensure the driving agent is able to **stay on the road**, without falling off.

### Instructions

1. **Review** the provided parameter descriptions.

2. Below, you'll find a summary of each tunable **reward function parameter**.

   • These parameters control the agent's learning and behavior.

3. **Modify the parameter values** in the reward function text file.

   • Adjust values to encourage the agent to complete the goal properly rather than remaining close to it.

4. Be **mindful of potential side effects** (e.g., excessive jumping or erratic behavior).

5. **Do not tune parameters** that are not explicitly mentioned in the *Problem* Section. For instance, if the *Problem* description is talking about making the agents collide less, but does not refer hitting the fence, **do not** change the fence collision penalty. It's very important that you can distill only the necessary adjustments, and only interfere with those. This also means that, depending on the *Problem* description, you might have to go against intuition, so if the problem mentions that it does want the agents to collide, you could consider attaching a positive value to the penalty, instead of the intuitive negative value associated with it.

6. **Output only** the modified reward function text file.

- The format of the output **must be identical** to the `.txt` file. For instance, don't forget to add the '*' symbol in between the scaling factor and the name of the rewards.

- **DO NOT** include explanations, additional comments, or any output other than the modified '.txt' file.

**Reward Parameter Descriptions**

**speedDriveReward**
- **Purpose**: Adjusts the reward based on the driving agent's speed and acceleration. It promotes aggressive yet controlled driving by rewarding speeds above a threshold and positive acceleration, while penalizing speeds that are too low or deceleration.

- **Adjustment Strategy**:
  - Use a positive value when encouraging the agent to drive at higher speeds with proper acceleration (e.g., exceeding a speed threshold like 22 f/s).
  - Decrease (or use a negative value) if a more conservative driving style is needed.
  - Tweak the magnitude to balance the benefits of speed with the risks of instability or inefficient acceleration.

**offRoadPenalty**
- **Purpose**: Penalizes the agent for leaving the designated drivable area. This negative reward enforces staying on track and promotes safe, controlled driving behavior.

- **Adjustment Strategy**:
  - Increase the absolute value (i.e., make it more negative) when staying on the road is critical.
  - Decrease or set to `0` if off-road behavior is acceptable or not a concern.
  - Note that this penalty is often amplified (e.g., multiplied in the termination call) to sharply discourage off-track behavior.

**lateralBiasReward**
- **Purpose**: Provides an additional reward (or penalty) based on the agent's lateral positioning relative to a desired trajectory or lane. It encourages maintaining consistent lateral alignment.

- **Adjustment Strategy**:
  - Values within the interval `[-1 , 1]` are optimized for the agent to stay within the track. If necessary to go off road (and only then) adjust this value to stay outside of the range mentioned.
  - The values are normalized in a way that, e.g., using `-0.5` as a value will make the car learn how to drive on the center of the left lane (idem for right side). Using higher absolute values lead to more extreme learning, so use carefully.
  - Increase with a positive value when the agent should favor a specific lateral alignment (for example, staying centered in a lane).
  - Use a negative value if deviations from a predefined safe or optimal path are undesirable.
  - Adjust the magnitude to control how strongly lateral positioning impacts the overall reward. If the values are too close to the extremes of the window, e.g. `-1` or `1`, the car will learn to drive

on the very edge of the road, so adjust this parameter with that in mind.

**stayOnTrackReward**
- **Purpose**: Reinforces the agent's ability to remain aligned and follow the designated track. It rewards the agent for correctly orienting itself toward the current track point and staying on the intended path.

- **Adjustment Strategy**:

  – Increase when the driving agent needs to follow a certain path, e.g. staying on the left/right side. This reward directly encourages the agent to drive alongside a certain spline define by the `lateralBiasReward`

  – The reward is applied in a graded manner based on the angle between the agent's forward direction and the direction to the current track point. Smaller angles (e.g., <10°) receive a higher positive contribution, while larger angles result in a reduction or small penalty.

  – Decrease or set to 0 if maintaining track alignment is not a priority in the scenario.

**Provided File: `reward_params.txt`**

```
reward = 0.1*speedDriveReward + -0.1*offRoadPenalty + 0.1*lateralBiasReward
    + 0.1*stayOnTrackReward
```

## B  Feedback Reward-Tuning Prompt with Statistics

In this section we show an example of a prompt containing the statistics and metrics collected by the agent. In this prompt, we ask the LM to refine the reward weights to better align the behavior of the agent to the desired preference. The example shows the prompt at the final iteration.

**LM Prompt**

### Iteration Update for Reward Function Tuning

Here's the reward functions you previously generated throughout consecutive iterations, alongside the respective summary of collected statistics from 5 different seeds. Note that the last iteration presented is the latest one you produced.

**Iteration 1**

**Reward Function:**

```
reward = 1.0*speedDriveReward + -1.0*offRoadPenalty + 0.0*lateralBiasReward
    + 0.0*stayOnTrackReward
```

**Statistics (summary across 5 seeds):**
- Total Episodes: 50.00
- Successful Episodes: $0.00 \pm 0.00$
- Success Rate (%): $0.00 \pm 0.00$
- Off Road Episodes: $3.00 \pm 2.74$
- Off Road Rate (%): $6.00 \pm 5.48$
- Timeout Episodes: $47.00 \pm 2.74$
- Timeout Rate (%): $94.00 \pm 5.48$

- Avg Speed Off Road (km/h): $18.63 \pm 8.12$

**Iteration 2**

**Reward Function:**

```
reward = 1.0*speedDriveReward + -1.0*offRoadPenalty + 0.0*lateralBiasReward
    + 1.0*stayOnTrackReward
```

**Statistics (summary across 5 seeds):**
- Total Episodes: 50.00

- Successful Episodes: $6.80 \pm 11.48$

- Success Rate (%): $13.60 \pm 22.95$

- Off Road Episodes: $41.60 \pm 11.26$

- Off Road Rate (%): $83.20 \pm 22.52$

- Timeout Episodes: $1.60 \pm 2.51$

- Timeout Rate (%): $3.20 \pm 5.02$

- Avg Speed Off Road (km/h): $72.18 \pm 31.66$

**Iteration 3**

**Reward Function:**

```
reward = 1.0*speedDriveReward + -10.0*offRoadPenalty + 0.0*lateralBiasReward
    + 1.0*stayOnTrackReward
```

**Statistics (summary across 5 seeds):**
- Total Episodes: 50.00

- Successful Episodes: $33.00 \pm 13.47$

- Success Rate (%): $66.00 \pm 26.94$

- Off Road Episodes: $14.60 \pm 10.85$

- Off Road Rate (%): $29.20 \pm 21.71$

- Timeout Episodes: $2.40 \pm 3.36$

- Timeout Rate (%): $4.80 \pm 6.72$

- Avg Speed Off Road (km/h): $109.70 \pm 10.10$

**Iteration 4**

**Reward Function:**

```
reward = 1.0*speedDriveReward + -20.0*offRoadPenalty + 0.0*lateralBiasReward
    + 1.0*stayOnTrackReward
```

**Statistics (summary across 5 seeds):**
- Total Episodes: 50.00

- Successful Episodes: $33.00 \pm 13.47$

- Success Rate (%): $66.00 \pm 26.94$

- Off Road Episodes: $14.60 \pm 10.85$

- Off Road Rate (%): $29.20 \pm 21.71$

- Timeout Episodes: $2.40 \pm 3.36$

- Timeout Rate (%): $4.80 \pm 6.72$

- Avg Speed Off Road (km/h): $109.70 \pm 10.10$

**Iteration 5 (Run A)**

**Reward Function:**

```
reward = 1.0*speedDriveReward + -50.0*offRoadPenalty + 0.0*lateralBiasReward
    + 1.0*stayOnTrackReward
```

**Statistics (summary across 5 seeds):**

- Total Episodes: 50.00

- Successful Episodes: $46.80 \pm 6.06$

- Success Rate (%): $93.60 \pm 12.12$

- Off Road Episodes: $3.20 \pm 6.06$

- Off Road Rate (%): $6.40 \pm 12.12$

- Timeout Episodes: $0.00 \pm 0.00$

- Timeout Rate (%): $0.00 \pm 0.00$

- Avg Speed Off Road (km/h): $127.06 \pm 4.03$

- Avg Speed Success (km/h): $136.22 \pm 6.32$

- Avg Steps Success: $849.87 \pm 66.85$

**Iteration 5 (Run B)**

**Reward Function:**

```
reward = 1.0*speedDriveReward + -100.0*offRoadPenalty +
    0.0*lateralBiasReward + 1.0*stayOnTrackReward
```

**Statistics (summary across 5 seeds):**

- Total Episodes: $50.00 \pm 0.00$

- Successful Episodes: $13.00 \pm 21.68$

- Success Rate (%): $26.00 \pm 43.36$

- Off Road Episodes: $11.00 \pm 9.27$

- Off Road Rate (%): $22.00 \pm 18.55$

- Timeout Episodes: $26.00 \pm 15.60$

- Timeout Rate (%): $52.00 \pm 31.21$

- Avg Speed Success (km/h): $126.67 \pm 17.31$

- Avg Speed Off Road (km/h): $47.55 \pm 31.86$

- Avg Steps Success: $1001.66 \pm 257.76$

**Notes:**

- **StepCount**: Represents the total number of simulation steps in that episode.

- **Avg_speed_kmh**: The average speed measured in km/h.

- **Outcome**: An episode is determined as

  - *OffRoad* if the cumulative off-road counter increased.

  - *Timeout* if the agent didn't complete the episode within max time steps.

  - *Successful* if a full lap was completed.

**Task:**

1. **Analyze** the simulation statistics in comparison with the intended behaviors described in the original problem:

- The agent should reach the goal as fast as possible, ie. in the least amount of time steps.

- The agent should not go off road, ie. we want to have an agent able to consistently meet the end goal.

- Keep in mind that, since we're in an RL scenario, it's not straightforward that reducing an explicit reward will produce the desired behavior (e.g. reducing speed might involve boosting/reducing other rewards). The decision must be made by accounting for all of the parameters.

2. **Identify** which aspects of your previous reward function require adjustment based on the performance metrics and any observed side effects. For instance, pay attention to the change in a given statistic from the previous iteration to the current, and see if the the previous change you applied has **improved** the performance or not. If it didn't, find a way to circle back to a more successful run and try a different approach.

3. **Refine** the scaling factors in the reward function accordingly. Try to be creative with the solution, ie. you don't need to go from 0.5 to 0.5, for instance, and can try different decimal points.

4. **Output** only the updated reward function file in the exact same format as before (do not include any additional commentary or explanation).

Your output should be the updated reward function file only.

## C  Hyperparameters and Training Setup

Table 2 reports the list of hyperparameters used for RL optimization. To train the agents, we used Unity ML-Agents framework (Juliani et al., 2018) and Proximal Policy Optimization (PPO) as optimization algorithm (Schulman et al., 2017).

| Parameter | Value |
|---|---:|
| Batch size | 1024 |
| Buffer size | 20480 |
| Learning rate | $3 \times 10^{-4}$ |
| $\beta$ | $5 \times 10^{-3}$ |
| $\epsilon$ | 0.2 |
| $\lambda$ | 0.95 |
| $\gamma$ | 0.99 |
| Time horizon | 64 |
| Number of epochs | 5 |
| MLP size | 5 layers |
| Hidden units per layer | 128 |

Table 2: The most important hyper-parameters of our approach and their respective values.

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
