# OpenReview forum: "Self-correcting Reward Shaping via Language Models for Reinforcement Learning Agents in Games"
_rl-conference.cc/RLC/2025/Workshop/RLVG — RLVG Workshop - RLC 2025_

### Official Review · Reviewer_r7ip · 2025-06-15
**Interesting problem, lacks enough experimnets**

**Rating:** 3
**Confidence:** 4

**Summary:**

Aiming to eliminate the need for expert reward engineering when new content or mechanics are introduced in video game environments, the author proposes an automatic reward tuning approach. This approach leverages a large language model to tune the weights of the reward model based on user-defined language-based behavioral goals. The language model in each iteration returns a weight vector for the reward model given 3 input sources: the description of the task in text format, the past history of weight vectors, and the agent's performance statistics from previous evaluations.

**Strengths:**

- It targets an important challenge in the adoption of RL in industry. The approach provides a human-in-the-loop setup, enabling game designers and developers with little reinforcement learning (RL) knowledge to provide more efficient feedback loops for the RL agents using language models.
- Good analysis on the relevant metrics and comparison with human expert

**Weaknesses:**

- Results are presented on only one test environment!
- Lack of proper baselines
- Although this method eliminates the need for reward tuning under minor changes in environment tasks, it seems to require defining a set of environment-specific feature functions when the environment's dynamics change significantly. So, human effort is not fully eliminated!
- No results or discussion on alternative RL method (e.g. an off-policy method).

**Best Paper Nomination:**

No

**Claims:**

Claims are supported by evidence provided, but empirical evidence is somewhat limited.

**Suggestions:**

- Consider presenting results on more environments and providing analysis on the reusability of the tuned LM from one environment to another!
- “ Overall, the LM makes relatively small but consistent adjustments across iterations”, Figure 3  does not support this ?! Please phrase the paragraph better.

---

### Official Review · Reviewer_Mrgo · 2025-06-16
**Interesting idea although narrow scope**

**Rating:** 3
**Confidence:** 4

**Summary:**

The paper proposes an automated approach for iteratively fine-tuning an RL agent’s reward function weights (assuming a linear reward function with interpretable weights), based on a user-defined language based behavioral goal. The approach involves using a LLM to propose adjustments to these weights given 1) a high-level description of the task and environment, (2) the full history of weight vectors, and (3) concise performance statistics from prior agent evaluations. These new weights are then used for RL training and evaluation and the process is repeated for a fixed number of iterations or until the desired performance is reached. The paper demonstrates that the automated LLM-based approach is competitive with a human expert without manual tuning.

**Strengths:**

- Direction - Automating reward function engineering with LLMs is an important and timely idea to the application of RL in video games.
- Method - The approach is novel and sensible.
- Writing - The paper is well-written.

**Weaknesses:**

- Scope - Approach is limited to environments where the reward function is already defined as a linear combination of interpretable components.
- Additional complexity - Idea introduces the need for manual prompt engineering and performance statistics feature engineering in place of reward engineering.
- Experiments - Only one environment and pre-defined reward function considered.

**Best Paper Nomination:**

No

**Claims:**

Yes, the claims are supported in the provided experiments for the chosen use-case (although are not convincingly demonstrated in general).

**Suggestions:**

- Experiments - Applying the idea to additional environments and reward functions would help to demonstrate whether the method can be more broadly applied outside of this single use-case.
- Scope - Additionally, applying the LLM at a lower level, or the inclusion of a VLM as suggested in the further work may be more general.
- Analysis - Additional analysis of the number of iterations would be interesting, i.e. rather than running for a fixed T=5 iterations, allowing the pipeline to run for longer to understand whether it is able to reach stable convergence would be insightful.

---

### Decision · Program_Chairs · 2025-06-19

**Decision:**

Accept

**Comment:**

The authors propose an automated reward tuning method to remove the need for expert reward engineering when game content or mechanics change. This method uses a large language model to iteratively adjust reward weights based on a textual task description, past weight configurations, and the agent’s previous performance data. This paper is well-written and presents an important contribution to the adaptation of RL in video games, and is definitely relevant to the topic of the workshop.
The reviewers point out the need for a more accurate phrasing of the claims and contributions based on the presented evidence.
We strongly encourage the authors to address these points in the camera-ready version.